# Ceftazidime–Avibactam Resistance in Carbapenem-Resistant *Klebsiella pneumoniae* Bloodstream Infections: Risk Factors and Clinical Outcomes

**DOI:** 10.3390/antibiotics14111085

**Published:** 2025-10-28

**Authors:** Ayten Yanık, Ömer Karaşahin

**Affiliations:** Department of Enfectious Disease, Erzurum Regional Training and Research Hospital, 25070 Erzurum, Türkiye; mrkrshn@gmail.com

**Keywords:** *Klebsiella pneumoniae*, ceftazidime–avibactam, carbapenem resistance, antibiotic resistance, mortality

## Abstract

Background/Introduction: Carbapenem-resistant Klebsiella pneumoniae (CRKP) bacteremia is a serious public health problem due to its high mortality rate and limited treatment options. This study aimed to identify risk factors associated with ceftazidime–avibactam (CAZ-AVI) resistance in CRKP bacteremia and to evaluate its impact on clinical outcomes. Methods: This retrospective single-center cohort study included adult patients with CRKP bloodstream infections treated at a tertiary hospital in Türkiye between January 2021 and December 2024. Demographic, clinical, and laboratory data were collected, and risk factors for CAZ-AVI resistance and 30-day mortality were analyzed. Results: Among 154 patients, 42.8% had CAZ-AVI-resistant strains. Resistant infections were associated with longer hospital stays and higher Charlson Comorbidity Index (CCI) scores. The resistance rate was lower in patients with intra-abdominal infections, while fluoroquinolone and fosfomycin use was more common in the resistant group. The overall 30-day mortality rate was 57%. Pitt bacteremia score and creatinine levels were identified as independent predictors of mortality. Discussion: CAZ-AVI resistance in CRKP bacteremia appears to develop in patients with prolonged hospitalization and high comorbidity burden. These factors likely increase exposure to resistant microorganisms and antibiotic pressure, complicating treatment outcomes. Conclusions: CAZ-AVI resistance in CRKP bacteremia is associated with specific clinical risk profiles and contributes to high mortality. Identifying high-risk patients and optimizing antimicrobial stewardship are essential to improve prognosis.

## 1. Introduction

Antimicrobial resistance is recognized as an important public health problem worldwide because it complicates the spread and treatment of infections and increases the risk of serious illness and death. Among multidrug-resistant microorganisms, carbapenem-resistant *Enterobacteriaceae* (CRE) is a particular concern, with high rates of antibiotic resistance and limited treatment options [1]. According to World Health Organization data, CRE strains cause the highest global disease burden among multidrug-resistant Gram-negative bacteria and rank first on the list of critical priority pathogens [2]. The most common CRE species is carbapenem-resistant *Klebsiella pneumoniae* (CRKP), and CRKP infections are associated with high morbidity, mortality, and economic burden. There is an urgent need to expand the limited treatment possibilities in order to reduce the burden of these infections [2,3].

Carbapenem resistance in *K. pneumonia* is gained through carbapenemase enzymes, which are usually found in the periplasmic space and inactivate β-lactam antibiotics via hydrolysis. The most common carbapenemases include *K. pneumoniae* carbapenemase (KPC), metallo-β-lactamases (NDM, VIM, IMP), and oxacillinases like OXA-48. Next-generation β-lactam/β-lactamase inhibitor combinations that target these enzymes have been developed [4]. Ceftazidime–avibactam (CAZ-AVI) is one such combination that is effective against carbapenemases such as KPC and OXA-48 but not against metallo-β-lactamases [5]. However, CRKP isolates may develop resistance to CAZ-AVI over time due to mutations at the active site of the blaKPC gene (e.g., the Ω-loop) and changes in ompK35/36 porins [6]. In 2015, the first CAZ-AVI-resistant CRKP strain isolated from a patient with no prior CAZ-AVI use was reported. Shortly after being introduced into clinical use, CAZ-AVI resistance rates increased to over 20% in some centers, and resistance continues to increase worldwide [7]. In Turkey, where OXA-48-type carbapenemases are endemic, CAZ-AVI is frequently used as a first-line treatment for CRKP infections [8]. This epidemiological context underscores the importance of CAZ-AVI but also raises concerns as resistance to this drug emerges.

Increased CAZ-AVI resistance rates necessitate a better understanding of the clinical features, risk factors, and consequences of CAZ-AVI-resistant CRKP infections. Certain risk factors for CAZ-AVI resistance have been identified in the literature, including previous CAZ-AVI use, renal replacement therapy, and concomitant lung infections [9,10,11,12,13]. However, clinical studies and data on the treatment of CAZ-AVI-resistant *K. pneumoniae* infections are still limited, and the clinical outcomes of these treatments remain unclear. Mortality rates in CAZ-AVI-resistant *K. pneumoniae* infections are also reported to be quite high [14]. Therefore, our aim in this study was to determine the risk factors associated with CAZ-AVI resistance in CRKP bacteremia and to evaluate the impact of this resistance on clinical outcomes.

## 2. Results

### 2.1. Cohort Characteristics

Of the 154 patients included in this study, 66 (42.8%) were diagnosed with CAZ-AVI-resistant and 88 (57.2%) with CAZ-AVI-susceptible *K. pneumoniae* bacteremia (Figure 1). The median age of the patients was 73 years (IQR: 63–80 years) and 86 (55.8%) were male.No statistically significant difference was found between gender and CAZ-AVI resistance (*p* > 0.05).

### 2.2. Antibiotic Susceptibility Results

The in vitro antimicrobial susceptibility results of the CAZ-AVI-resistant and -susceptible *K. pneumoniae* strains are summarized in Figure 2. Rates of resistance to agents such as piperacillin-tazobactam, cefepime, ceftazidime, meropenem, imipenem, levofloxacin, and trimethoprim–sulfamethoxazole were over 70% in both groups. Although rates of amikacin and colistin resistance were lower in the CAZ-AVI-susceptible group, the differences were not statistically significant (*p* =0.177 and *p* = 0.352, respectively).

### 2.3. Risk Factors for CAZ-AVI Resistance

The demographic and clinical characteristics of the patients according to their CAZ-AVI susceptibility results are presented in Table 1. Patients infected with CAZ-AVI-resistant strains tended to be older and male, but these differences were not statistically significant. CAZ-AVI resistance was significantly more common in patients with a high Charlson Comorbidity Index (CCI) and with prolonged hospitalization prior to CRKP infection. In contrast, the resistance rate was significantly lower among patients with concomitant intra-abdominal infection (Table 1). In addition, CAZ-AVI resistance was significantly more frequent in patients who used fosfomycin or fluoroquinolone prior to blood culture (Table A1).

### 2.4. Multivariate Analysis Results

According to the logistic regression analysis, longer hospital stay before CRKP infection (OR: 1.011; 95% CI: 1.004–1.018; *p* =0.003) and high CCI (OR: 1.271; 95% CI: 1.102–1.466; *p* =0.001) were independent risk factors for CAZ-AVI resistance (Table 2). A 1-day increase in hospital stay was associated with 1.1% higher odds and a 1-point increase in CCI with 27.1% higher odds of CAZ-AVI resistance. The ROC curves for CCI and length of hospital stay prior to CRKP infection to differentiate CAZ-AVI resistance are presented in Figure A1. The power of CCI to predict CAZ-AVI resistance (area under the ROC curve) was 0.637 (95% CI: 0.549–0.726; *p* = 0.004) and that of length of hospital stay prior to CRKP infection was 0.724 (95% CI: 0.644–0.804; *p*< 0.001). CCI ≥6.5 predicted CAZ-AVI resistance with 59.1% sensitivity and 61.4% specificity, while a hospital stay ≥39.5 days predicted resistance with 75.8% sensitivity and 60.2% specificity.

### 2.5. Mortality and Associated Factors

The demographic and clinical characteristics of the patients according to 30-day mortality are presented in Table 3. The 30-day mortality rate was 57.6% (n = 38) in the CAZ-AVI-resistant group and 56.8% (n = 50) in the CAZ-AVI-susceptible group (*p* = 0.925). In the univariate analysis, central venous catheterization, mechanical ventilation, and nasogastric catheterization were significantly more frequent among patients with 30-day mortality (*p* = 0.009, *p* = 0.013, *p* = 0.001). Concomitant lung infection was also associated with higher mortality (*p* = 0.018). In laboratory findings, patients with 30-day mortality had significantly higher white blood cell count, neutrophil count, creatinine, urea, AST, and total bilirubin levels and significantly lower platelet count (*p* = 0.009, *p* = 0.001, *p* = 0.001, *p* = 0.012, *p* = 0.028, *p* = 0.001). INCREMENT-CPE, Pitt bacteremia score, and qSOFA values, which are used as clinical severity scores, were significantly higher among nonsurvivors (*p* < 0.001, *p* < 0.001, *p* = 0.003) (Table 3).

Inappropriate empirical and targeted therapies included the use of antibiotics with no in vitro activity against CRKP isolates, such as carbapenems (meropenem, imipenem), extended-spectrum cephalosporins (ceftriaxone, cefepime), piperacillin–tazobactam, fluoroquinolones (ciprofloxacin, levofloxacin), and aminoglycosides when resistance was documented.No significant difference in mortality was observed between patients who received targeted and empirical antibiotic therapy or according to antibiotic subgroup (Table A2).

### 2.6. Multivariate Analysis Results (Mortality)

Variables significant in univariate analysis were included in the multivariate logistic regression model. According to multivariate logistic regression analysis, each unit increase in creatinine level nearly doubled the odds of mortality (OR: 1.95; 95% CI: 1.09–3.50; *p* = 0.025), while each point increase in Pitt bacteremia score quadrupled the odds of mortality (OR: 4.02; 95% CI: 1.53–10.55; *p* = 0.005) (Table 4). The distribution of Pitt bacteremia scores according to 30-day mortality is presented in Figure 3A, and the ROC curves of Pitt bacteremia score and creatinine for 30-day mortality are presented in Figure 3B. The area under the ROC curve for Pitt bacteremia score as a predictor of 30-day mortality was 0.814 (95% CI: 0.742–0.886; *p* < 0.001). At a cut-off value of 4.5, Pitt bacteremia score predicted mortality in CAV-AVI-resistant patients with 95.5% sensitivity and 57.6% specificity. The area under the ROC curve of mortality prediction for creatinine was 0.704 (95% CI: 0.621–0.787; *p* < 0.001). With a cut-off value of 1.01 mg/dL, creatinine predicted mortality in CAV-AVI-resistant patients with 61.4% sensitivity and 78.8% specificity.

In the Kaplan–Meier analysis, 30-day mortality was significantly higher among patients with CAZ-AVI-resistant CRKP bacteremia who had a Pitt bacteremia score of 4.5 or higher and creatinine levels of 1.01 or higher compared to those with a Pitt bacteremia score of 4.5 and creatinine below 1.01 (Figure 4). A description of the experimental results, their interpretation, and the experimental conclusions that can be drawn are provided.

## 3. Discussion

In this study, CAZ-AVI resistance in CRKP bacteremia was found to be alarmingly prevalent and associated with identifiable risk factors, yet it did not significantly increase short term mortality. To our knowledge, this work is among the few studies that specifically focus on CAZ AVI resistance in bloodstream infections caused by CRKP.

The development of CAZ AVI resistance was strongly associated with prolonged hospitalization and a high comorbidity burden. Patients who had longer hospital stays prior to infection and higher Charlson Comorbidity Index (CCI) scores were at greater risk of yielding CAZ AVI-resistant isolates. These findings are consistent with earlier reports indicating that extended hospitalization creates a favorable environment for the selection of resistant strains. Campogiani etal. showed that most *Klebsiella pneumoniae* isolates that developed CAZ AVI resistance were recovered from patients with lengthy hospital stays and extensive antibiotic exposure, suggesting that resistance is facilitated by antibiotic pressure [10]. Similarly, a study from an oncology center identified prolonged hospitalization as the sole independent predictor of CAZ AVI resistance among CRKP isolates [15]. In our cohort, an inpatient stay of approximately 39 days or longer prior to infection emerged as a clinically meaningful threshold for resistance risk, in line with those observations.

Patients with greater comorbidity (higher CCI scores) were also at increased risk for CAZ AVI-resistant infections. Individuals with multiple underlying conditions are more frequently exposed to invasive procedures, treated with broad spectrum antibiotics, and may have compromised immune defenses; these factors facilitate colonization and subsequent infection with resistant organisms [13]. Indeed, patients with multiple comorbidities and prolonged hospital stays are thought to carry resistant microorganisms in the gastrointestinal flora and to be more likely to develop infections with these strains [16]. In our multivariable analysis, CCI remained an independent predictor; each one-point increase in CCI was associated with an approximately 27% increase in the odds of resistance.

Another important observation was the relationship between prior antibiotic exposure and CAZ AVI resistance. Prior use of certain agents—particularly fluoroquinolones or fosfomycin—was significantly more frequent among patients with CAZ AVI-resistant infections. Although these variables did not remain independent predictors in multivariable modeling, their univariable associations suggest that suboptimal regimens may contribute to the selection of resistant subpopulations. Ahmed etal. reported that CAZ AVI resistance occurred more often in patients with a history of meropenem use, positing that exposure to broad spectrum antibiotics can select for mutants harboring resistance mechanisms [9]. In our study, meropenem exposure did not discriminate between groups, likely because nearly all patients had already received a carbapenem. However, CAZ AVI resistance was significantly more common among those who had received fluoroquinolone or fosfomycin before bacteremia. This pattern may reflect suppression of susceptible bacteria with outgrowth of resistant clones when these agents are insufficiently effective, or adaptive mutations triggered by antibiotic stress. Even though these agents did not retain independent significance, we believe these patterns warrant careful attention in clinical practice.

None of the patients in our cohort received empiric CAZ AVI before culture results were available. This reflects national antibiotic stewardship policies in Türkiye that restrict the use of newer agents unless susceptibility is documented. Consequently, prior CAZ AVI exposure could not be evaluated as a risk factor in our cohort. However, this circumstance suggests that the observed resistance likely developed de novo under other antibiotic pressures or disseminated within the hospital environment. A study from Italy documented clonal spread of CAZ AVI-resistant strains even among patients without a history of CAZ AVI use [17], indicating that resistance can emerge and propagate independently of direct drug exposure.

An intriguing protective factor was the presence of concomitant intra-abdominal infection, which appeared to be associated with a lower likelihood of CAZ AVI resistance. Most such cases in our study involved cholangiosepsis following endoscopic retrograde cholangiopancreatography (ERCP). These patients generally had fewer comorbidities and shorter hospital stays, thereby exhibiting fewer risk factors for resistance development. In a previous series of 46 CRKP infections of intra-abdominal origin, no CAZ AVI resistance was reported [10].This finding suggests that certain infection foci may inherently carry a lower risk for resistance development and that the epidemiology of CAZ AVI resistance may vary by the site of infection.

Despite marked differences in risk factors and patient profiles, CAZ AVI resistance was not associated with short term mortality in our study. Thirty-day mortality in the resistant group (57.6%) was very similar to that in the susceptible group (56.8%). This finding aligns with other reports that did not detect a significant mortality difference between CAZ AVI susceptible and resistant CRKP infections [14]. A plausible explanation is that, when appropriate alternative therapies are employed, mortality becomes more dependent on host factors and illness severity than on the resistance profile per se. In our cohort, most patients with CAZ AVI susceptible infections ultimately received CAZ AVI, whereas resistant infections were typically treated with colistin based combinations. Comparable mortality rates suggest that the disadvantages conferred by resistance may be partially offset when aggressive treatment and source control are achieved.

Nevertheless, overall mortality in our cohort was high (57%). In the literature, 30-day mortality for CAZ AVI-resistant *K. pneumoniae* infections ranges from 16% to 41% [9,18,19,20]. Two factors likely account for our higher rate: first, we included only bloodstream infections, which have substantially higher mortality than other foci. In the meta-analysis by Hu etal., mortality averaged ~54% in CRKP bacteremia compared with ~14% in urinary tract infections caused by the same pathogen [21]. Second, in the study reporting the lowest mortality (16.2%), approximately 60% of isolates regained carbapenem susceptibility during treatment, enabling early transition to effective therapy [18]. In our cohort, no such scenario was observed; all cases involved highly resistant organisms from the outset. These data underscore that CRKP bacteremia carries a high risk of death regardless of CAZ AVI resistance status.

The principal determinants of mortality in our study were indicators of critical illness and organ dysfunction. The Pitt bacteremia score is widely used to predict prognosis in bloodstream infections; a high score (≥4 points) reflects severe initial illness and has been strongly linked to mortality in multiple studies [18,19]. In our analysis, patients with a Pitt score ≥4.5 had a significantly higher likelihood of death. Because the score reflects severe sepsis features (e.g., hypotension, altered mental status, need for intensive care), higher scores independently increase the risk of mortality [22]. Another independent predictor in CAZ AVI-resistant infections was renal dysfunction. In particular, acute kidney injury (AKI) during the course of infection significantly increased mortality risk. Boattini et al. reported that a substantial proportion of patients who died from CAZ AVI-resistant *K. pneumoniae* bacteremia developed AKI [18]. Generally, elevated serum creatinine in severe sepsis signals a poor prognosis [23]. AKI not only reflects severe hypotension and impaired tissue perfusion but also complicates therapy: antibiotic doses are often reduced, potentially diminishing efficacy in critical infections. Indeed, reduced CAZ AVI dosing in patients with CRE infections has been associated with higher mortality [24]. Thus, the creatinine elevation observed in our study—reflecting renal injury—was an independent risk factor for death both because it marks severe illness and because it constrains treatment.

This study has several strengths. First of all, it is one of few studies to focus on CAZ-AVI resistance in CRKP bacteremia and examined national data in the light of the international literature. Both microbiological and clinical risk factors were assessed in detail using a comprehensive data set. Real-life data on patient management (empirical and definitive treatments, clinical course) were examined and the outcomes of CAZ-AVI resistance were addressed using a holistic approach. However, our study has some limitations. Due to the retrospective design, the data were collected retrospectively, and there is a risk of incomplete or biased information for some parameters. The fact that it was performed in a single center may limit the generalizability of the results. The widespread presence of OXA-48 and NDM enzymes, which are major contributors to carbapenem resistance in Türkiye, suggests that our findings may differ in regions with distinct resistance profiles. Indeed, previous studies have reported that OXA-48–like carbapenemases are endemic in Mediterranean countries, whereas KPC-type enzymes are more prevalent in the United States. Therefore, local resistance patterns should be taken into account when generalizing our results to other geographic settings. In particular, very few patients in our study had a history of CAZ-AVI use because of national health policies, and therefore it was not possible to fully evaluate the impact of this variable on resistance development. Additionally, molecular typing could not be performed for all isolates and specific resistance mechanisms could not be examined. Finally, treatment approaches are not standardized and varied according to the clinical discretion of the physician, which may result in confounding factors in outcomes due to some patients receiving different treatments.

## 4. Materials and Methods

### 4.1. Study Design and Patient Selection

This single-center, retrospective cohort study included patients who had CRKP-positive blood cultures in a tertiary hospital between January 2021 and December 2024. Infection control committee records from this period were screened to identify patients with *K. pneumonia* isolates, and those confirmed to be carbapenem resistant were included in the study. Only the first isolate was evaluated for each patient to avoid data redundancy due to repeated sampling.

Ethics committee approval was obtained from the local ethics committee of the institution where the study was conducted (date: 9 April 2025, decision No. 93). Patient consent was not required due to the retrospective study design.

### 4.2. Data Collection and Analysis

Patient data were collected retrospectively through the infection control committee records and hospital information management system. Demographic characteristics, concomitant diseases, history of invasive interventions, intensive care admission, previous antibiotic use, and foci of infection were recorded.We specifically recorded whether the initial empirical antibiotic therapy was appropriate or inappropriate. Empirical therapy was defined as appropriate if it included at least one antibiotic active against the CRKP isolate (based on in vitro susceptibility results), and inappropriate if it lacked any active agent against the pathogen. Inappropriate targeted therapy was defined as any definitive regimen initiated after availability of culture and susceptibility results that did not include at least one in vitro active agent against the isolate or failed to provide adequate exposure.The day of CRKP infection was defined as the specimen collection date of the first CRKP-positive blood culture. The pre-infection length of stay was calculated as the number of days from the date of hospital admission to that specimen collection date. Thirty-day mortality was defined as death occurring within 30 days of the CRKP infection date. Primary outcome measures were 30-day mortality and clinical recovery. Risk factors were identified by first comparing CAZ-AVI-resistant and -susceptible groups in terms of clinical and demographic variables, then including significant variables in multivariate logistic regression analysis. Factors associated with mortality were evaluated separately, and the prognostic values of clinical indices such as Pitt bacteremia score, qSOFA, and INCREMENT-CPE score were analyzed.

### 4.3. Inclusion and Exclusion Criteria

The study included all patients aged 18 years and older who had CRKP-positive blood cultures at our hospital between 2021 and 2024. All 154 patients included in the study were diagnosed with a hospital-acquired bloodstream infection. Patients who were under 18 years of age, had polymicrobial bacteremia or recurrent CRKP bacteremia within 30 days of the index event, or died before receiving targeted treatment based on culture results were not included in the study.

### 4.4. Microorganism Identification

Venous blood was inoculated into BD BACTEC™ Plus Aerobic/F and incubated in the BD BACTEC™ FX automated blood culture system (Becton Dickinson, Franklin Lakes, NJ, USA) according to the manufacturer’s instructions. Bottles flagged positive by the instrument were Gram-stained and subcultured onto appropriate media for isolation. Species identification and antimicrobial susceptibility testing of *K. pneumoniae* isolates obtained from blood cultures were performed using the BD Phoenix™ automation system (Becton Dickinson, USA). Susceptibility tests were interpreted according to the clinical breakpoints of the European Union Committee on Antimicrobial Susceptibility Testing (EUCAST). CAZ-AVI resistance was defined as a minimum inhibitor concentration (MIC) >8/4 µg/mL according to EUCAST. Carbapenem resistance was defined as nonsusceptibility to imipenem, meropenem, or ertapenem.

### 4.5. Statistical Analysis

Data analysis was performed using IBM SPSS Statistics software 26.0 (IBM Corp., Armonk, NY, USA). To determine the risk factors for CAZ-AVI resistance, we first compared patients with CAZ-AVI-resistant and -susceptible infections. Categorical data were analyzed by chi-square or Fisher’s test; continuous data were analyzed using the Mann–Whitney U test. Variables found to be significant in this analysis were included in the multivariate logistic regression model to identify independent predictors. In the mortality analysis, factors affecting 30-day mortality were first examined by univariate (log-rank and Cox regression) and then with multivariate logistic regression analyses. Receiver operating characteristic (ROC) curve analysis was performed for independent variables. In all tests, *p* < 0.05 was accepted as the criterion for statistical significance. Analyses were performed with SPSS 26.0 software.

## 5. Conclusions

This study showed that CAZ-AVI resistance is alarmingly prevalent among patients with CRKP bacteremia, especially those with a high comorbidity burden and prolonged hospitalization. The risk of developing resistance was also increased in patients under high antibiotic pressure and exposed to invasive interventions. CAZ-AVI resistance clinically complicates the treatment process. However, in our study it did not have a direct impact on mortality; the patient’s general clinical condition was a more significant factor. In this context, Pitt bacteremia score and renal dysfunction (creatinine level in particular) were the main predictors of prognosis. Resistance development can be reduced and patient management improved through early risk assessment, careful planning of empirical therapy, and enhanced infection control measures. Prospective and multicenter studies are still needed to increase our knowledge on this topic.

## Figures and Tables

**Figure 1 antibiotics-14-01085-f001:**
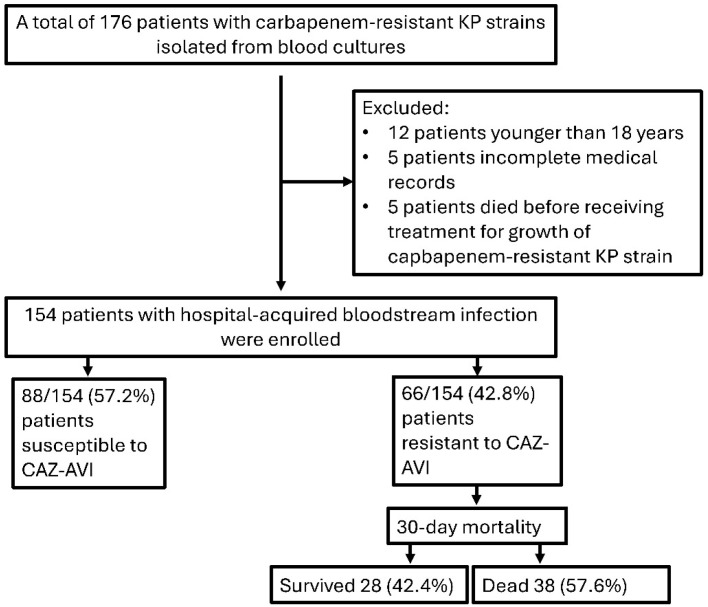
Flow chart of patients included in this study. (KP: *Klebsiella pneumoniae*, CAZ-AVI: Ceftazidime–avibactam).

**Figure 2 antibiotics-14-01085-f002:**
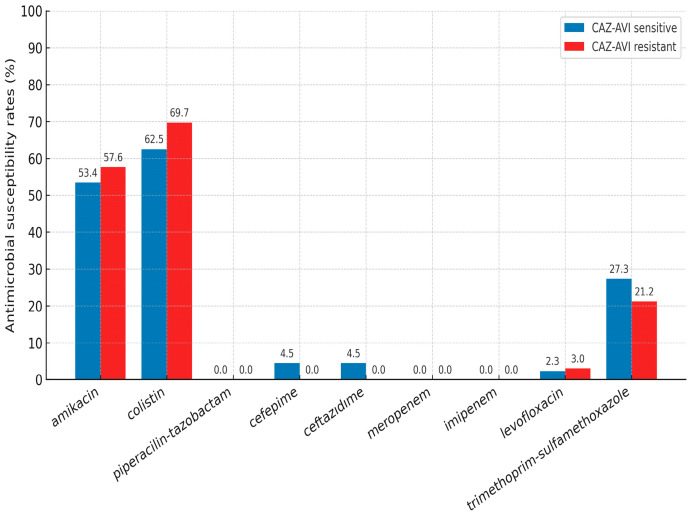
Antimicrobial susceptibility results of *K. pneumoniae* strains (CAZ-AVI: Ceftazidime–avibactam).

**Figure 3 antibiotics-14-01085-f003:**
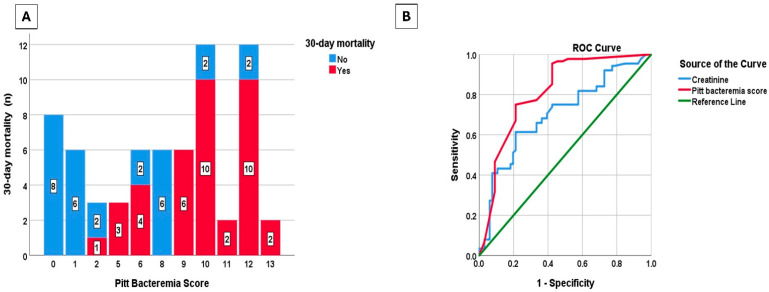
Distribution of Pitt bacteremia score by 30-day mortality (**A**), receiver operating characteristic (ROC) curves of Pitt bacteremia score and creatinine for the prediction of 30-day mortality (**B**).

**Figure 4 antibiotics-14-01085-f004:**
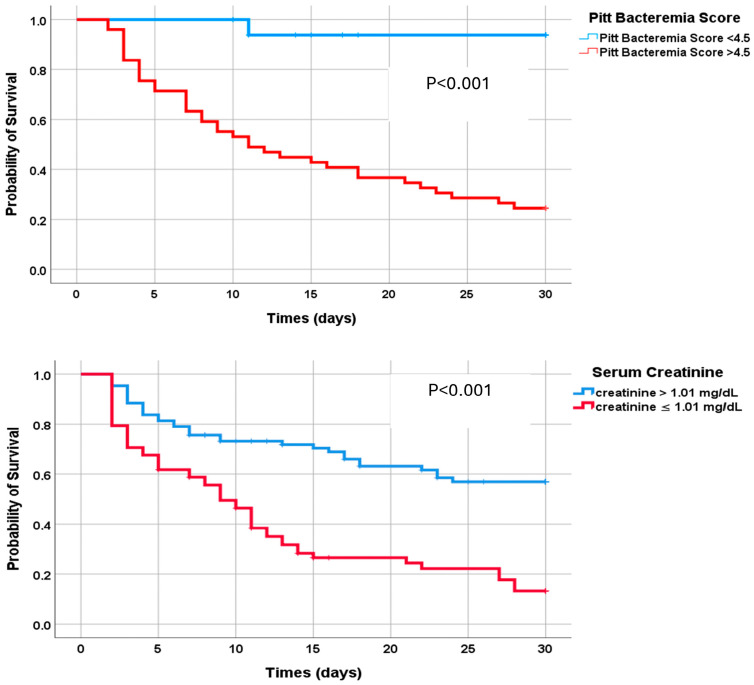
Kaplan–Meier analysis to compare 30-day survival rates for patients with Pitt bacteremia scores above and below 4.5 and creatinine levels above and below 1.01.

**Table 1 antibiotics-14-01085-t001:** Demographic and clinical characteristics of the patients according to CAZ-AVI susceptibility results.

	Resistant (n = 66)	Susceptible (n = 88)	*p*
*Demographics*			
Age (y), median (IQR)	73 (66–80)	70 (63–82)	0.591
Gender (male), n (%)	42 (63.6)	44 (50.0)	0.092
*Comorbidities, n* (%)			
Diabetes mellitus	16 (24.2)	22 (25.0)	0.914
Hypertension	33 (50.0)	36 (40.9)	0.262
COPD	13 (19.7)	20 (22.7)	0.650
Chronic kidney disease	12 (18.2)	13 (14.8)	0.570
Coronary artery disease	25 (21.2)	18 (20.5)	0.909
Chronic heart disease	10 (15.2)	6 (6.8)	0.093
Cerebrovascular disease	20 (30.3)	17 (19.3)	0.114
Immunosuppression	16 (24.2)	19 (21.6)	0.698
Malignancy	16 (24.2)	23 (26.1)	0.789
*Concomitant infection, n* (%)			
Lung	30 (45.5)	34 (38.6)	0.396
Intra-abdominal	10 (15.2)	32 (36.4)	**0.003**
Catheter-related bloodstream	18 (27.3)	18 (20.5)	0.322
Urinary tract	8 (12.1)	7 (8.0)	0.388
Central nervous system	-	-	-
*Invasive medical interventions in last month, n* (%)			
Central venous catheterization	56 (84.8)	64 (72.7)	0.073
Mechanical ventilation	52 (78.8)	60 (68.2)	0.144
Renal replacement therapy	18 (27.3)	21 (23.9)	0.630
Nasogastric catheterization	51 (77.3)	54 (61.4)	0.055
Urinary catheterization	57 (86.4)	73 (83.0)	0.564
Surgery	18 (27.3)	36 (40.9)	0.079
*Laboratory result, median (IQR)*			
WBC count (×10^9^/L)	11.12 (7.11–17.56)	11.28 (8.83–17.21)	0.305
Hemoglobin (g/dL)	8.90 (8.38–9.90)	9.25 (8.85–9.58)	0.055
Neutrophil count (10^9^/L)	8.99 (5.25–15.99)	9.12 (6.77–14.56)	0.535
Lymphocyte count (10^9^/L)	0.97 (0.40–1.44)	0.82 (0.47–1.69)	0.901
Platelet count (×10^9^/L)	194 (126–332)	147 (100–266)	0.082
Albumin (g/L)	26 (23–30)	26 (23–30)	0.816
Blood urea nitrogen (mg/dL)	35 (24–73)	39 (21–60)	0.379
Creatinine (mg/dL)	0.96 (0.68–1.81)	0.92 (0.56–1.56)	0.645
Aspartate aminotransferase (U/L)	28 (16–56)	40 (23–76)	0.057
Alanine aminotransferase (U/L)	29 (19–61)	35 (23–75)	0.357
Total bilirubin (mg/dL)	0.69 (0.31–1.80)	0.80 (0.59–2.07)	0.153
C-reactive protein (mg/L)	179 (134–268)	178 (127–235)	0.141
*Clinical severity at infection onset, mean ± SD*			
INCREMENT-CPE	8.18 ± 4.13	8.36 ± 5.64	0.658
Pitt bacteremia score	7.23 ± 4.36	7.25 ± 4.72	0.816
qSOFA	1.90 ± 0.87	1.76 ± 1.14	0.715
Charlson Comorbidity Index	7.23 ± 3.04	5.74 ± 2.74	**0.003**
*Antimicrobial therapy, n* (%)			
Inappropriate targeted therapy	24 (36.4)	27 (30.7)	0.458
Inappropriate empirical therapy	56 (84.8)	71 (80.7)	0.501
30-day mortality, n (%)	38 (57.6)	50 (56.8)	0.925
Length of hospital stay before CRKP infection (days), median (IQR)	62 (38–141)	30 (13–56)	**<0.001**

COPD: Chronic obstructive pulmonary disease, CRKP: Carbapenem-resistant *Klebsiella pneumonia*, IQR: Interquartile range.

**Table 2 antibiotics-14-01085-t002:** Multivariate Analysis of Risk Factors for CAZ-AVI-Resistant *Klebsiella pneumoniae* Bacteremia.

Variables	Univariate		Multivariate	
	OR (95% CI)	*p*	OR (95% CI)	*p*
Intra-abdominal infection	0.31 (0.14–0.70)	**0.004**	0.67 (0.25–1.84)	0.442
Charlson Comorbidity Index	1.20 (1.07–1.35)	**0.003**	1.271 (1.102–1.466)	**0.001**
Prior quinolone use	2.51 (1.25–5.03)	**0.010**	1.76 (0.77–4.03)	0.180
Prior fosfomycin use	4.67 (1.43–15.22)	**0.011**	1.90 (0.50–7.19)	0.345
Length of hospital stay prior to CRKP infection (days)	1.010 (1.004–1.016)	**0.001**	1.011 (1.004–1.018)	**0.003**

CRKP: Carbapenem-resistant *Klebsiella pneumonia.*

**Table 3 antibiotics-14-01085-t003:** Demographic and clinical characteristics of patients according to 30-day mortality.

	Survivors (n = 28)	Nonsurvivors (n = 38)	*p*
*Demographics*			
Age (y), median (IQR)	71 (66–79)	76 (66–80)	0.298
Gender (male), n (%)	18 (64.3)	24 (63.2)	0.925
*Comorbidities, n* (%)			
Diabetes mellitus	8 (28.6)	8 (21.1)	0.481
Hypertension	13 (846.4)	20 (52.6)	0.618
COPD	3 (10.7)	10 (26.3)	0.115
Chronic kidney disease	6 (21.4)	6 (15.8)	0.557
Coronary artery disease	6 (21.4)	8 (21.1)	0.971
Chronic heart disease	4 (14.3)	6 (15.8)	0.866
Cerebrovascular disease	10 (35.7)	10 (26.3)	0.412
Immunosuppression	4 (14.3)	12 (31.6)	0.105
Malignancy	4 (14.3)	12 (31.6)	0.105
*Concomitant infection, n* (%)			
Lung	8 (28.6)	22 (57.9)	**0.018**
Intra-abdominal	6 (21.4)	4 (10.5)	0.222
Catheter-related bloodstream	9 (32.1)	6 (15.8)	0.117
Urinary tract	2 (7.1)	6 (15.8)	0.287
Central nervous system	-	-	-
*Invasive medical interventions in last month, n* (%)			
Central venous catheterization	20 (71.4)	36(94.7)	**0.009**
Mechanical ventilation	18 (64.3)	34 (89.5)	**0.013**
Renal replacement therapy	6 (21.4)	12 (31.6)	0.360
Nasogastric catheterization	16 (57.1)	35 (92.1)	**0.001**
Urinary catheterization	22 (78.6)	35 (92.1)	0.113
Surgery	6 (21.4)	12 (31.6)	0.360
*Laboratory result, median (IQR)*			
WBC count (×10^9^/L)	8.64 (6.54–11.55)	13.82 (8.23–18.17)	**0.009**
Hemoglobin (g/dL)	9.20 (8.40–10.40)	8.75 (8.28–9.80)	0.725
Neutrophil count (10^9^/L)	6.84 (4.49–10.72)	10.89 (7.47–17.79)	**0.001**
Lymphocyte count (10^9^/L)	1.01 (0.58–1.42)	0.90 (0.32–1.55)	0.604
Platelet count (×10^9^/L)	224 (183–337)	158 (97–332)	**0.036**
Albumin (g/L)	26 (22–31)	26 (23–28)	0.917
Blood urea nitrogen (mg/dL)	25 (20–47)	38 (30–75)	**0.012**
Creatinine (mg/dL)	0.70 (0.43–0.96)	1.25 (0.82–2.11)	**0.001**
Aspartate aminotransferase (U/L)	22 (16–47)	45 (18–106)	**0.028**
Alanine aminotransferase (U/L)	25 (19–44)	40 (19–152)	0.115
Total bilirubin (mg/dL)	0.49 (0.30–1.15)	1.60 (0.58–2.74)	**0.001**
C-reactive protein (mg/L)	239 (144–276)	168 (118–255)	0.099
*Clinical severity at infection onset, mean ± SD*			
INCREMENT-CPE	5.79 ± 3.73	9.95 ± 3.48	**<0.001**
Pitt bacteremia score	4.07 ± 4.27	9.55 ± 2.66	**<0.001**
qSOFA	1.50 ± 1.00	2.21 ± 0.62	**0.003**
Charlson Comorbidity Index	7.11 ± 2.69	7.32 ± 3.30	0.845
*Antimicrobial therapy, n* (%)			
Inappropriate targeted therapy	12 (42.9)	12 (31.6)	0.347
Inappropriate empirical therapy	23 (82.1)	33 (86.8)	0.425

COPD: Chronic obstructive pulmonary disease, WBC: White blood cell, SD: Standart deviation, qSOFA: Quick sequential organ failure assessment, IQR: Interquartile range.

**Table 4 antibiotics-14-01085-t004:** Multivariate Analysis of Risk Factors for 30-Day Mortality.

Variables	Univariate		Multivariate	
	OR (95% CI)	*p*	OR (95% CI)	*p*
Lung	2.92 (1.47–5.79)	**0.002**	1.95 (0.77–4.97)	0.154
Central venous catheterization	7.20 (1.39–37.23)	**0.019**	0.53 (0.07–4.31)	0.553
Mechanical ventilation	4.72 (1.29–17.20)	**0.019**	0.26 (0.04–1.63)	0.153
Nasogastric catheterization	8.75 (2.17–35.36)	**0.002**	5.59 (1.02–30.65)	0.061
WBC count (×10^9^/L)	1.12 (1.02–1.23)	**0.014**	1.04 (0.83–1.30)	0.731
Neutrophil count (10^9^/L)	1.14 (1.04–1.25)	**0.007**	1.05 (0.83–1.32)	0.676
Platelet count (×10^9^/L)	0.97 (0.93–1.001)	0.112		
Blood urea nitrogen (mg/dL)	1.02 (0.99–1.04)	0.107		
Creatinine (mg/dL)	2.80 (1.22–6.40)	**0.015**	1.95 (1.09–3.50)	**0.025**
Aspartate aminotransferase (U/L)	1.01 (1.00–1.02)	0.052		
Total biluribine (mg/dL)	2.85 (1.41–5.75)	**0.003**	1.16 (0.87–1.55)	0.295
INCREMENT-CPE	1.35 (1.15–1.58)	**<0.001**	0.94 (0.79–1.12)	0.499
Pitt bacteremia score	1.46 (1.23–1.74)	**<0.001**	1.38 (1.11–1.71)	**0.003**
qSOFA	2.99 (1.46–6.14)	**0.003**	1.33 (0.48–3.68)	0.587

qSOFA: Quick sequential organ failure assessment.

## Data Availability

The raw data supporting the conclusions of this article will be made available by the authors on request.

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
