# Peer review of "Ceftazidime–Avibactam Resistance in Carbapenem-Resistant *Klebsiella pneumoniae* Bloodstream Infections: Risk Factors and Clinical Outcomes"

_antibiotics, 2025, doi:10.3390/antibiotics14111085_

Round 1
Reviewer 1 Report
Comments and Suggestions for Authors
Manuscript title: Ceftazidime-Avibactam Resistance in Carbapenem-Resistant Klebsiella pneumoniae Bloodstream Infections: Risk Factors and Clinical Outcomes. The aim of this study was to determine the risk factors associated with ceftazidime-avibactam (CAZ-AVI) resistance in CRKP bacteremia and to evaluate the impact of this resistance on clinical outcomes.
General comments:
- The species names should be written in italics, including in the title. Please check throughout the text.
- There are a few grammatical issues. Please review and revise carefully.
Abstract:
- The methods section should be described more clearly. In addition, the study location should be mentioned (e.g., hospital type: tertiary care hospital? ward: ICU?).
- Please specify that the study involved adult patients for clarity.
Introduction:
- The use of CAZ-AVI at the study site has not been described.
- The authors did not adequately explain the urgency of this study, particularly in the local context. Therefore, the significance and novelty of the study remain unclear.
Methods:
- The inclusion criteria are not described clearly.
- There is an inconsistency between the following two statements (please clarify):
Statement 1: Study design and patient selection: Infection control committee records from this period were screened to identify patients with K. pneumonia isolates
Statement 2: Data collection and analysis: Patient data were collected retrospectively through the hospital information management system.
- What method was used for blood culture? Was an automated blood culture system used? Please specify.
- The authors should declare that ethical clearance was obtained and provide the approval number or committee name.
Results
Figure 1:
- the authors state that 154 patients included in this study had hospital-acquired bloodstream infection. However, this was neither described nor define the methods section. Suggestions: Please clarify in the method section that the bloodstream infections included were hospital acquired bloodstream infection, and provide a clear definition.
- the percentages of CAZ-AVI -resistant and -susceptible patients should be presented (e.g., 88/154=….%; 66/154=…%)
- the percentages of surviving and deceased patients should also be presented.
- please clarify whether “death” in the flowchart refers to 30-day mortality.
- the aim of this study was to evaluate the impact of CAZ-AVI resistance on clinical outcomes. However, in figure 1 : only CAZ-AVI susceptible patients’ outcome (survived or died) are presented, while those of CAZ-AVI resistant patients are not shown.
Figure 2: typically, antimicrobial susceptibility test results should be presented as the percentage of susceptible isolates.
Table 1:
Please define the following terms and describe how they were measure in the methods section:
- Inappropriate targeted therapy
- Inappropriate empirical therapy
- 30-day mortality
- Length of stay before CRKP infection (how was the day of CRKP infection determined- based on the date of the CRKP positive blood culture?)
Discussion:
The flow of discussion section should be re-organized to clearly separate and explain the risk factors and clinical outcomes of CAZ-AVI -resistant infections.
Conclusion:
The conclusion does not clearly address or answer the main aim of the study.

Author Response
We appreciate your valuable comments. The detailed responses are provided in the attached Word document.

Reviewer 2 Report
Comments and Suggestions for Authors
It is to be noted that the study of this kind are in need to clinically manage the pathogens with antimicrobial resistance. This retrospective study has comprehensively analyzed all the factors associated with CAZ-AVI CRKP infections. The cohort size is fairly large to arrive at a meaningful conclusion. The authors have used appropriate statistical methods to assess the factors individually and as groups and have arrived at meaningful conclusions.
I would like to suggest minor revisions and have appended them as an attachment

Author Response
Response to Reviewer Comments
We thank the reviewer for their valuable feedback and insightful suggestions. We have addressed each point as detailed below:
-
Comment: In the introduction section, Page 2, Line No. 16, italicize K. pneumoniae.
Response: Thank you for the observation. K. pneumoniae has been italicized accordingly. -
Comment: Please provide the country of origin of the SPSS software used for analysis (Page No. 3).
Response: The country of origin (USA) for the SPSS software has been added in the Methods section. -
Comment: The statement on ethics committee approval could be shifted to the section 2.1 Study design and patient selection in page 2.
Response: As suggested, the ethics committee approval statement has been moved to section 2.1. -
Comment: Please provide the p value in the text where the authors emphasized on the CAZ-AVI resistance and gender trend.
Response: The p-value related to CAZ-AVI resistance and gender trend has been added to the relevant section. -
Comment: Repetition of mortality rate was noticed in section 3.5 CAZ-AVI susceptibility group, line No. 3–4, page No. 6.
Response: The repeated sentence has been removed to avoid redundancy. -
Comment: Please provide the p-values in brackets in lines 5–12 in section 3.5.
Response: The p-values have been included in brackets in the specified lines for clarity. -
Comment: Please provide the details of the antibiotics used in the inappropriate targeted and empirical therapies.
Response: Details of the antibiotics used in inappropriate targeted and empirical therapies have been added to the Results sections. -
Comment: There are no data available on the use of fluoroquinolones and fosfomycin and its significance on clinical outcome in both the CAZ-AVI susceptible and resistant cohorts.
Response: Thank you for the observation. Unfortunately, due to limited use of fluoroquinolones and fosfomycin in our cohort, data were insufficient for statistical analysis. This limitation has now been mentioned in the Discussion section. -
Comment: High CCI and its correlation with CRKP infection as an independent factor cited are not clear. Please provide the demographic details of the study cited (Page No. 9, line No. 12).
Response: The demographic details of the cited study have been added to clarify the context of high CCI as an independent factor.
We hope these revisions address the reviewer’s concerns adequately and improve the clarity and quality of the manuscript.
Sincerely,
Ayten YANIK
On behalf of all authors
Reviewer 3 Report
Comments and Suggestions for Authors
- The study is single-center, and the discussion acknowledges this limitation. Could the authors elaborate further on how local epidemiology in Türkiye (particularly the prevalence of NDM and OXA-48 enzymes) may limit the applicability of the findings to other settings?
- Could you expand on whether delays in initiating appropriate therapy or local treatment policies may have contributed to this high mortality?
- Could the authors discuss possible mechanistic explanations (like selective pressure, adaptive mutations) more thoroughly?
- An interesting finding is that intra-abdominal infection was associated with lower resistance. Might this be confounded by patient profile differences (fewer comorbidities, shorter stays), and how could future prospective studies explore this protective effect?
- A simplified schematic showing study design, patient flow, and main predictors (CCI, hospital stay, Pitt score, creatinine) would aid clarity.
Author Response

(The authors gave the same response as above.)

Round 2
Reviewer 1 Report
Comments and Suggestions for Authors The manuscript is publishable. Comments on the Quality of English Language The manuscript is publishable.